# A Review of the Impact of Maternal Prenatal Stress on Offspring Microbiota and Metabolites

**DOI:** 10.3390/metabo13040535

**Published:** 2023-04-09

**Authors:** Venkata Yeramilli, Riadh Cheddadi, Juhi Shah, Kyle Brawner, Colin Martin

**Affiliations:** 1Division of Pediatric Surgery, Department of Surgery, University of Alabama at Birmingham, Birmingham, AL 35294, USA; 2Burnett School of Medicine, Texas Christian University, Fort Worth, TX 76129, USA; 3Department of Biology, Lipscomb University, Nashville, TN 37204, USA

**Keywords:** stress, microbiome, short chain fatty acids, intestine, butyrate

## Abstract

Maternal prenatal stress exposure affects the development of offspring. We searched for articles in the PubMed database and reviewed the evidence for how prenatal stress alters the composition of the microbiome, the production of microbial-derived metabolites, and regulates microbiome-induced behavioral changes in the offspring. The gut–brain signaling axis has gained considerable attention in recent years and provides insights into the microbial dysfunction in several metabolic disorders. Here, we reviewed evidence from human studies and animal models to discuss how maternal stress can modulate the offspring microbiome. We will discuss how probiotic supplementation has a profound effect on the stress response, the production of short chain fatty acids (SCFAs), and how psychobiotics are emerging as novel therapeutic targets. Finally, we highlight the potential molecular mechanisms by which the effects of stress are transmitted to the offspring and discuss how the mitigation of early-life stress as a risk factor can improve the birth outcomes.

## 1. Increasing Prevalence of Anxiety and Stress

Anxiety and stress disorders are a major public health concern. In addition, social determinants of health, provide additional stressors for at risk populations [1]. Furthermore, the effects of the COVID-19 pandemic and social isolation have had profound effects collectively on societal mental health [2]. These factors are a heterogenous cluster of mental health disorders that can manifest common physical symptoms seen with anxiety (i.e., tachycardia, dyspnea, or gastrointestinal distress) [3]. Other clinical symptoms associated with anxiety disorder include worry, fear, or avoidance that is excessive or out of proportion to the situation, persistent, and associated with impairments in social, occupational, or other important areas of functioning [3]. Anxiety disorders have worsened in the United States with the prevalence of anxiety increasing approximately 6.7% from 2008 to 2018 amongst 18–25-year-olds [4,5]. The increasing prevalence of anxiety and stress in society should be considered as a precursor to several co-occurring mental health problems, cardiovascular disease, and metabolic syndrome [4,6,7]. In addition, because of the overall increase in stress and anxiety, more attention should be given to the effects of stress on health and non-neurological diseases for the individual and infants born to parents in stressful environments or pre-existing mental health conditions.

## 2. Effect of Stress on Microbiota Composition

It is well-established that stress alters the gut microbiome. Our lab and others have shown that psychological stress experienced by pregnant rodents leads to gut dysbiosis in the offspring [8,9,10]. The offspring likely inherit the dysbiotic microbiota from their mothers because psychological stress during pregnancy alters the composition of the maternal gastrointestinal and vaginal microbiota, and substantial overlap exists between the structure of the bacterial communities at these sites with the structure of the gut bacterial community of the offspring [11]. Maternal psychological stress during pregnancy is also associated with offspring gut dysbiosis in humans [12]. In addition to inheriting a dysbiotic maternal microbiota, prenatal stress might contribute to dysbiosis in the offspring by inducing changes in the expression of genes in the fetal intestine related to inflammation and the recruitment of immune cells [8]. Interestingly, the stress-induced dysbiotic maternal vaginal microbiota was found to partially contribute to gene expression changes in the paraventricular nucleus of the hypothalamus of male offspring following adulthood stress [8]. Furthermore, male adult offspring of prenatally stressed mothers produce more corticosterone in response to a stressful stimulus than adult males born from non-stressed mothers [9]. These results suggest that the transmission of stress-induced maternal dysbiotic microbiota to newborns continues to shape the stress responses of the offspring into adulthood through dysregulation of the hypothalamic pituitary adrenal (HPA) axis.

Other studies have investigated the effect of juvenile and adulthood stress on the microbiota independent of prenatal stress. Mice exposed to immobilization stress (IS) show an increase in the relative abundance of *Proteobacteria* and *Escherichia coli* and a decrease in the relative abundance of lactobacilli in the gut. Importantly, the transplant of fecal microbiota from IS mice into specific pathogen free mice results in anxiety-like behavior in the recipients, which is associated with increased neuroinflammation, increased levels of blood pro-inflammatory cytokines, and a greater recruitment of monocytes to the colon compared with the recipients of fecal microbiota from non-stressed mice [13]. Additional studies that have subjected mice to other stressors have confirmed that stress tends to reduce lactobacilli levels in the gut [14,15,16]. Although lactobacilli have specifically received attention as being impacted by stress, it is also appreciated that stress impacts the microbiota composition at a community-wide level and not just certain taxa [17]. Mechanistically, stress hormones might directly act on microbes to facilitate their growth [18,19,20,21]. Furthermore, stress is known to produce many changes in gut physiology including a reduced production of gastric acid [22], the prevention of bile release from the gallbladder to small intestine [23], reduced small intestinal motility [24], and altered levels of secretory IgA [10,25,26]. Any of these changes to the gut physiology can alter the gut microbiota composition.

## 3. The Gut-Brain Axis and Microbial Metabolites

Anxiety and stress modulation of the brain–gut effect is well-elucidated [27,28] as brain–gut signaling can directly affect the microbiota via intestinal motility and the release of neurotransmitters [29]. Comparatively, gut–brain signaling has gained attention in recent years in fields investigating psychiatric, neurodevelopmental, and neurodegenerative pathologies [30,31]. Growing evidence suggests that microbiota dysfunction is a key element recurring in patients with neuropsychiatric disorders such as autism spectrum disorders and other comorbid behavioral disorders like anxiety, cognitive impairment, and depression [32,33]. Similarly, studies have shown that changes in the maternal microbiome during pregnancy because of antibiotics, use of probiotics, diet variation, and exposure to stress can modulate the neurodevelopment of the offspring [34,35].

The neurobiological interaction between the gut and brain is based on bidirectional signals between the metabolites produced by the gut bacteria and the nervous system [36]. Short chain fatty acids (SCFA) are key bacterial metabolites with regard to the gut microbiota signaling pathways. These are organic acids derived from the microbial fermentation of dietary fibers and starch in the intestines. While anaerobic fermentation is the largest source, SCFAs can also result from amino acid metabolism [37,38] and are found in high quantities in dairy products, hence, the SCFA concentration can vary strongly with diet [38]. In addition to their known roles as trophic factors and energy supply substrates to the gut [39], SCFAs are thought to play a major role in stress and anxiety modulation [28], and reduced levels of SCFAs have been found to be associated with chronic stress. Indeed, SCFAs are important regulators of enterochromaffin cell (ECC) functions by mediating the tryptophan metabolism [40], these enteroendocrine cells lining the gastrointestinal tract synthesize 95% of the body serotonin 5-hydroxytryptamine (5-HT) [41]. 5-HT released from the basal membrane of intestinal ECC cells then interacts with receptors from neurons in the enteric nervous system (ENS) to modulate motility. Vagal afferents signal to the nucleus of the solitary tract (NTS) and the dorsal raphe nucleus (DRN), the latter of which houses most of the brain’s 5-HT neurons. These areas then interact with emotion-regulating brain networks that influence mood and psychological stress [42].

It has been shown that chronic systemic inflammation, especially in the form of infections, is associated with behavioral alterations, as illustrated by the neuropsychiatric manifestations of COVID-19 [43]. Likewise, increased inflammation in the periphery and the brain can alter the blood–brain barrier permeability and neurotransmitter metabolism, therefore increasing adverse behavioral issues (e.g., stress, anxiety) and emotional disturbance [44]. SCFAs—butyrate, propionate, and acetate—are involved in inducing regulatory T cells (Tregs) and as a result, control inflammation [45]. SCFAs also promote the synthesis of IL-10, a major anti-inflammatory cytokine [43], thus accordingly, SCFAs seem to reduce neuroinflammation and exert a neuroprotective effect [46].

The existence of a complex bidirectional system between the microbiome and the brain offers promising strategies to treat both gastrointestinal and neuropsychiatric pathology. With the advancements in understanding the gut–brain axis physiology, new therapies have emerged. Probiotics have evolved into psychobiotics, as they exert anxiolytic and antidepressant effects [47]. Garrett et al. subjected infant rodents to early life stress and treated the mice with either a neutral substance (control), citalopram (SSRI), or *Bifidobacterium infantis* (psychobiotic). Surprisingly, the results were comparable between the psychobiotic and the citalopram group, suggesting that probiotics may have a broader clinical application than previously considered [48]. This novel class of therapeutics exerts its effect on two main pathways. First, systemic effects on the hypothalamic–pituitary–adrenal axis and glucocorticoid hemostasis. Several studies have elucidated a strong association between a depressive state and activation of the inflammatory response [49], in addition to an aberrant secretion of pro-inflammatory cytokines [50]. The second category of psychobiotic effects is via neurotransmitters and protein modulation, mainly γ-aminobutyric acid (GABA), glutamate, and brain-derived neurotrophic factor (BDNF) [47]. Other therapeutics involve trace elements such as iron, zinc, calcium, and magnesium. The latter was deemed as the most promising micronutrient that connects to the gut–brain axis [51]. Several studies have suggested that magnesium deficiency can cause not only dysbiosis, but also psychiatric symptoms. One study noted excessive anxiety and depressive-like behavior in the magnesium-deficient mice. Additionally, neuroinflammatory markers were increased [52]. These studies suggest a crucial effect of magnesium on the equilibrium of the gut–brain axis. Another study found a significant improvement in the quality of life in patients suffering from depression following treatment with probiotics and magnesium orotate [53]. Further research on such novel and multidisciplinary interventions may become useful as prophylactic or adjuvant therapies for brain–gut pathologies.

## 4. The Effect of Probiotics Supplementation on SCFA Production

Intestinal microorganisms significantly impact the metabolic processes in the body via the production of SCFAs. A proper balance of the microbiome is crucial to maintain optimal health and prevent many diseases. This balance can be achieved through probiotic supplementation. Several studies have been performed to assess the impact of deliberate reproduction or the administration of probiotics on the health of people who exhibit either a reduction or overabundance of certain groups of microorganisms. The effect of oral consumption of *Lactobacillus plantarum* P-8 for 4 weeks was tested on a cohort of 33 volunteers including young, middle-aged, and elderly participants. The authors found an increase in *Bifidobacterium* and other beneficial bacteria, whereas *Desulfovibrio* and other opportunistic pathogens were decreased following a 4-week supplementation. Furthermore, a significant increase in the acetate and propionate levels in all age groups was found, indicating that the SCFA levels can be altered via probiotic supplementation [54]. In another study, a long-term administration of probiotics for 15 weeks in mice increased the production of SCFAs and resulted in improved memory and learning abilities in a d-galactose-induced model of aging [55]. Butyrate production was significantly increased in 24 people following the consumption of fermented salami, along with probiotic *Lactobacillus rhamnosus* HN001 and citrus fiber for 4 weeks, indicating that dietary intervention with fermentable fibers can modulate the production of SCFAs [56]. Colorectal cancer is strongly correlated with decreased levels of SCFAs and microbiome dysbiosis [57]. Therefore, several clinical trials and in vitro studies have been conducted to understand the role of probiotic-induced SCFA production on tumor progression and the inhibition of cancer cell proliferation. In a mouse model of colon cancer, the administration of *Butyrivibrio fibrisolvens* MDT-1, known for its production of butyrate, inhibited the progression of tumor development [58]. In a human cancer cell line (Caco-2 cells), probiotics reduced cell proliferation via an increase in the production of SCFAs and increased the adhesion of probiotic bacteria to cancer cells [59]. Moens et al. assessed the ability of an aqueous probiotic suspension (Symprove^TM^) to influence the human gut using an in vitro gut model (M-SHIME^®^ system). Following dosing with Symprove over a 3-week period and following the colonization and growth of these probiotics in the tissues, the authors observed higher proximal and distal colonic lactate concentrations. The lactate stimulated growth of lactate-consuming bacteria resulted in increased SCFA production, especially butyrate [60]. In a clinical trial, 10 colorectal cancer patients receiving *Lactobacillus gasseri* OLL271 6:LG21 for 12 weeks showed an increase in the number of *Lactobacillus* spp. and a decrease in *Clostridium perfringens* in the colon. They also exhibited an increase of isobutyric acid in the feces and increased NK cell activity, indicating the possibility of preventing colorectal carcinogenesis with probiotics [61]. Individuals with obesity had an altered composition of gut microbiome that led to changes in the activity and function of several metabolic parameters including an altered SCFA profile. Improvements in microbial dysbiosis, gut function, and obesity have been reported following the use of probiotics. For example, the administration of *Bifidobacterium pseudocatenulatum* CECT 7765 to rats on a high-fat diet increased the intestinal barrier function, reduced endotoxemia, and accelerated metabolism [62]. In another study, feces from 21 overweight and 19 normal weight school-age students were examined for changes in the microbial and SCFA profiles following the consumption of a probiotic drink containing *Lactobacillus casie*. A higher concentration of butyric and propionic acids was found in the stool of overweight children compared to normal-weight samples. There was a marked improvement in the profile of intestinal microbiota and an increase in the concentration of acetic acid in the stool of children with obesity when *Lactobacillus casei* was administered for a longer duration of 6 months [63,64]. In other studies, the intake of probiotics such as fermented goat’s milk improved glycemic control in people with type 2 diabetes, and these changes were associated with changes in the concentrations of inflammatory cytokines and acetic acid [65]. To investigate the role of probiotics on cardiometabolic disorders induced by a high-fat diet, Cavalcante et al. administered *Lactobacillus fermentum* to rats for 4 weeks and observed an improvement in the cardiovascular and biochemical parameters. The increased production of SCFAs is thought to modulate vasodilation and induce hypotension in this model as a potential mechanism of action [66]. The role of intestinal microbiome in brain disorders is poorly understood. In the feces of children with autism spectrum disorders (ASD), significantly lower levels of SCFAs were found compared to healthy children. However, the profiles of these SCFAs improved after children with ASD were administered probiotics [67]. Taken together, research highlighting the immunomodulatory effects of SCFAs induced by probiotics (Table 1) in several disease models is promising, but their mechanisms of action still need further study to better understand their etiology and pathogenesis and to propose new therapies.

## 5. The Effect of Probiotic Supplementation on the Stress Response

Stress is a ubiquitous part of modern life. Chronic stress can lead to anxiety, depression, and several neurological disorders [68]. Psychobiotics, defined as microorganisms (probiotics) that confer mental health benefits to the host through interactions with the commensal gut bacteria, is emerging as a probiotic-based alternative therapy to treat certain neurological and neurodegenerative diseases [69]. The gut microbiota, which can be easily modified through combinations of diet and probiotics, is a promising target for protection against the harmful effects of stress on the gut–brain axis [70]. Sudo et al. observed an amplified corticosterone response following stress in germ-free mice. This response was normalized following supplementation with *Bifidobacteriul infantis*, indicating that the microbiome regulates the HPA axis and endocrine signaling [71]. Based on these findings, several groups have explored the potential of probiotics in either protecting or restoring the symptoms caused by chronic stress [47,72]. Animal studies have shown that probiotics reduce anxious behaviors in response to physical stress. For example, rats, when administered a probiotic formulation (PF) consisting of *Lactobacillus helveticus* R0052 and *Bifidobacterium longum* R0175 daily for 2 weeks, demonstrated less defensive probe burying in a conditioned defensive burying test, a screening model for anti-anxiety agents [73]. In other studies, experimental rats spent more time in an open field test after hypothermia following the administration of *L. rhamnosus* [74], and reduced the immobilization time in a forced swim-test after a 5-week supplementation with a probiotic mix [75]. In a clinical trial, volunteers participated in a double-blind, placebo-controlled, randomized study with a probiotic formulation administered for 30 d and assessed with the Hopkins Symptom Checklist (HSCL-90), the Hospital Anxiety and Depression Scale (HADS), and the Perceived Stress Scale. The daily subchronic administration of probiotics alleviated psychological distress in volunteers and reduced depressive symptoms [73]. Following treatment with either the *Lactobaccilus Plantarum* DR7 or P8 strain, adults experiencing moderate stress symptoms on the Perceived Stress Scale (PSS-10) at the baseline reported fewer of these symptoms by the end of the study [76]. In contrast, in two other clinical trials in healthy volunteers, no changes in the stress-related parameters were observed after either a 2- or 8-week supplementation of *Lactobacilli*, *Bifidobacteria*, and *Streptococci* strains [77,78], respectively. Thus, even though probiotics positively influence human stress-related outcomes, the effects are not always consistent, and more research is needed to understand the mechanism(s) through which probiotics exert their influence. The gut–brain axis, involving interactions between the vagus nerve, immune system, endocrine system, and the microbiome-induced metabolites and neurotransmitters such as dopamine, serotonin, histamine, and SCFAs, likely plays a role [79]. After a 12-week supplementation of *Lactobacillus plantarum* P-8 in a cohort of stressed adults, Ma et al. observed an alleviation of stress symptoms and an enhancement in the diversity of neurotransmitter synthesizing bacteria and a concomitant increase in the production of neuroactive metabolites [76,80]. A recent meta-analysis based on randomized controlled trials to understand the efficacy of probiotics on stress in healthy volunteers revealed that probiotics can reduce subjective stress levels in healthy volunteers and may alleviate stress-related anxiety/depression levels, but without any significant effect on the cortisol levels [81]. Taken together, *Lactobacillaceae* and *Leuconostocaceae* (as per the new taxonomic classification proposed by Zheng et al. [82]) occupy a significant part of the gut microbiota [83]. We highlight some of the therapeutic capabilities of lactobacilli-derived metabolites and their impact on the stress response (Table 2). While there is a strong correlation between probiotics and stress, more research is needed to understand how probiotic supplementation modulates the HPA axis and whether probiotics can be used as a therapy to manage the allostatic load, the cumulative burden of chronic stress and life events.

## 6. Father’s Role in Offspring Microbiome

The maternal–infant microbial relationship has been relatively well-studied. However, little is known about the role of the father in shaping the offspring microbial communities. Commensal bacteria play a significant role in modulating host physiology and behavior through the gut–brain-axis pathway [84,85]. Most studies have focused on microbial inheritance based on a variety of factors such as the type of birth (vaginal versus C-section) [86,87], feeding method (breast fed versus formula fed) [88,89], and skin-to-skin contact after birth [90,91]. Microbes can be transmitted horizontally from the father to mother as well as to the baby because cohabiting family members share microbiota with one another [92,93]. Interestingly, Yael Gurevich et al. proposed that microbes may play a key role in paternal care. Using a computational model, the authors demonstrated that paternal care was induced by the microbes rather than the father’s genetic make-up [94]. Thus, microbe-induced paternal care could modulate horizontal microbial transmission to the offspring. A recent study identified novel interactions between paternal diet, maternal care, and offspring microbiome and stress reactivity [95]. Paternal high-fat diet exposure significantly altered the adult offspring Fermicutes to Bacteroidetes ratio and behavior [95]. These findings are consistent with the emerging concept that stress leads to dramatic changes in the composition of gut bacteria and regulates stress-induced changes in social behavior. These changes in microbiota may play a mechanistic role in the generational effects of stress. Further studies, however, are required to investigate the direct effects of paternal psychological stress on offspring microbiota and if probiotic therapy is efficacious in reversing the effects of paternal stress on F1-generation offspring.

## 7. Mechanisms of Cross-Generational Transmission of Prenatal Stress

A growing body of research now supports the idea that offspring are affected by either prenatal or possibly preconception parental stress exposures [96] Here, we outline some of these mechanisms and discuss the concepts of intergenerational and transgenerational transmission of stress. For a more comprehensive review on epigenetic transmission of stress, see Mbiydzenyuy et al. [97] and Cao-Lei et al. [98].

Stress during pregnancy is associated with an increased risk for neurodevelopmental disorders [99]. Our lab and others have shown that stress during pregnancy impacts offspring microbial composition and intestinal function [8,10,11,100]. Several factors contribute to the stress-induced effects on the maternal milieu. One is the transfer of genes that are vulnerable to stress from mother to child [101]. Other mechanisms involve the direct impact of trauma and stress on the fetal and post-natal environment [102]. For example, the placental corticotrophin-releasing factor (CRF) regulates the fetal HPA axis and is required for the optimal production of glucocorticoids and androgens from the fetal adrenal gland [103,104]. Maternal stress negatively modulates fetal HPA axis feedback and development by increasing the placental CRF production and signaling [105]. The fetal HPA axis, even though fully developed by the third trimester, is still vulnerable and sensitive to environmental insults and stress [106,107,108]. Increased circulating glucocorticoids following prenatal stress exposes the fetus to excessive glucocorticoids and is thought to be one of the mechanisms by which maternal stress effects are passed to the offspring [109]. Maternal stress also promotes an inflammatory state in the placenta by increasing cytokine production. In stressed mice, elevated levels of IL-6 within the placenta desensitized the HPA axis in male offspring [110]. Glucocorticoids act as immune modulators by altering the migration, differentiation, and proliferation of thymocytes, monocytes, and neutrophils [109,111,112]. Consistent with this idea, Valeria et al. demonstrated a systemic reduction in B and T cells in the F1 and F2 offspring following prenatal stress in mice [113]. A clinical study also showed that gestational stress decreased lymphocyte activity [114]. Taken together, these studies indicate that stress-induced glucocorticoid activity modulates the maternal and offspring immune system. At a molecular level, prenatal stress alters placental function and signaling through changes in the epigenetic machinery. Gestational stress reduces the levels of 11β-hydroxysteroid dehydrogenase type 2 (11β-HSD2), an enzyme that protects the fetus by converting high levels of maternal cortisol into inactive cortisone [115]. This decrease in 11β-HSD2 is associated with an increase in 11β-HSD2 promoter methylation and the expression of DNA methyltransferase 3A in the placenta [116]. In human studies, increased methylation of 11β-HSD2 and Nr3c1, a gene encoding for glucocorticoid receptor (GR), in the placentas from mothers who reported anxiety and depression during pregnancy was positively correlated with impairments in infant neurobehavior [117,118]. Howerton et al. reported the expression of O-linked N-acetylglucosamine transferase (OGT), a nutrient sensing enzyme in the placenta following prenatal stress. Interestingly, the X-linked OGT expression was significantly decreased in male placentas compared to female placentas, indicating that OGT is a potential mediator of sex-specific programming in offspring [119]. Taken together, these studies demonstrate an epigentic link between prenatal stress exposure and programming in the placenta.

While most of the research on the effects of maternal stress has focused on understanding the placental environment, little is known about the mechanisms by which maternal preconception stress affects offspring development. Lineage studies in Holocaust survivors suggest that HPA stress axis sensitivity in offspring can be affected via alterations in DNA methylation of stress regulatory genes [120,121,122]. To investigate the effect of preconception stress on oocytes, Zaidan et al. subjected female rats to chronic unpredictable stress for 7 days and found that the oocyte expression of CRF was 18.5 times greater in stress-exposed females compared to the controls [123]. In support of this idea, another study found that maternal childhood abuse was correlated with greater placental CRF production during pregnancy [124]. These findings suggest that maternal preconception stress programs offspring development through epigenetic changes to the oocytes. Further studies are needed to understand how preconception stress alters the maternal biology and affects subsequent pregnancies and offspring development (Figure 1).

## 8. Non-Traditional Models to Investigate Stress–Microbiome–Behavior Relationships

Most studies employ rodents as a model to understand stress–microbiome–behavior relationships. In addition to mammalian models, we suggest using non-traditional model systems as a comparative approach to evaluate the stress–microbiome–behavior dynamics. Ki-Hyeon Seong et al. subjected male fruit flies to restraint stress and found that paternal restraint stress activated p38 in testicular germ cells and altered the transcriptome of drosophila offspring by modulating the changes in gene expression and metabolite content related to energy metabolism [125]. A recent study showed that specific gut bacteria regulate behavior in Drosophila. They found that germ-free males were less aggressive and mated inefficiently with female flies. This behavior, however, could be reversed by re-colonization with commensal bacteria [126]. Similar studies indicate that Drosophila can be used as a robust model to further understand the microbiome [127,128]. Teleost fish is emerging as another promising model to study the impact of early life stress on the microbiome and behavior [129,130]. Germ-free zebrafish exhibit altered locomotion and anxiety-like behavior. However, treatment with *Lactobacillus plantarum* was sufficient to attenuate anxiety-like behavior in conventionally raised zebrafish larvae [131]. Early-life stress leads to the development of anxiety-like behavior in adults, and these phenotypes are epigenetically transferred to the subsequent two generations [132]. Finally, *C. elegans* could be another candidate model to study the transgenerational effects of stress response because of their short life cycle. Exposure to early life environmental stressors in worms leads to increased longevity and resistance to such stressors later in life; phenotypes that are also passed on to subsequent generations [133,134,135]. Overall, these findings indicate that the transgenerational inheritance of stress responses is a more widely conserved role across a variety of animal models. In conclusion, a comparative approach employing both traditional and non-traditional animal models will help identify novel patterns or unique features that will enhance our understanding of these complex relationships between stress and the microbiome.

## 9. Discussion and Conclusions

In this review, we discussed how prenatal stress alters the microbiome and promotes anxiety-like behavior and described how probiotic supplementation has a profound effect on the stress response and SCFA production. Bidirectional gut–brain axis signaling is complex and regulates the microbial composition through the production of several metabolites and neurotransmitters [27,28,29].To utilize probiotics as candidate drugs, more studies are needed to understand the mechanisms through which probiotic supplementation modulates the stress response. Even though a strong correlation between prenatal maternal stress and hyperactivity of the HPA axis exists, the relationship between stress and fetal exposure to glucocorticoids is not relatively well-understood. In particular, other variables such as the type of stress (trauma, psychological, or physical) and timing of prenatal stress needs to be further investigated.

We highlight some of the epigenetic and non-genetic factors in the transmission of prenatal stress effects. With mounting evidence supporting the intergenerational transmission of stress from mother to offspring, it is important to also investigate the factors and ways to stop such transmissions. Cancer diagnosis and treatment is a stressful experience. Numerous stress management interventions have been reported to facilitate psychological and physiological adaptation and optimal health outcomes in cancer patients and survivors [136]. Similar stress management interventions during pregnancy should be employed to determine whether it will lead to improved birth outcomes.

Finally, in addition to cohort studies, we propose using a variety of animal models to complement our understanding of the molecular and cellular mechanism of stress and to develop therapies to reverse the adverse effects of stress. For example, due to federal laws [137,138], it is very difficult to obtain oocytes to study preconception stress. As a result, animal models will have to be used for the foreseeable future. In the Martin lab, we are developing zebrafish and *C. elegans* models to investigate the roles of early life and oxidative stress on the progeny and subsequent generations. In conclusion, because early-life stress is a risk factor, there should a greater awareness to focus on the health of both parents to achieve the most beneficial outcome for children.

## Figures and Tables

**Figure 1 metabolites-13-00535-f001:**
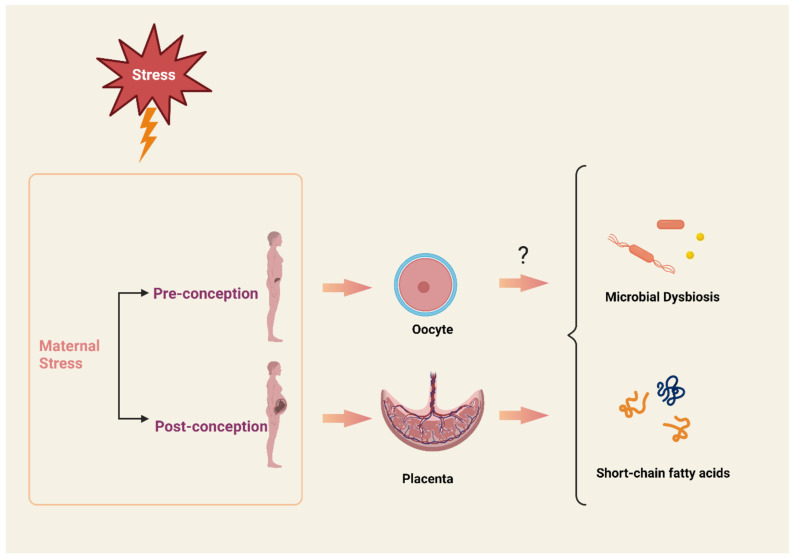
Cross-generational transmission of parental stress can impact offspring microbiome and metabolite production. Maternal stress during pre-conception or post-conception modulates the oocyte and placenta, respectively, leading to microbial dysbiosis. It is unclear how maternal preconception stress via the oocyte can influence the microbiome. Cartoon was created with BioRender.com.

**Table 1 metabolites-13-00535-t001:** The effect of probiotics on SCFA production.

Probiotic	Species	Effects (↑ = Increase ↓ = Decrease)	Reference
*Lactobacilus plantarum* P-8	Human	↑ *Bifidobacterium* ↓ *Desulfovibrio* ↑ Acetate & Propionate	[54]
Probiotic mixture containing *Lactobacillus paracasei* ssp. *paracasei* BCRC 12188, *Lactobacillus plantarum* BCRC 12251, and *Streptococcus thermophilus* BCRC 13869	Mice	↑ SCFAs ↑ Improved memory and learning ability	[55]
Fermented salami + *Lactobacillus rhamnous* + citrus fiber	Human	↑ Butyrate	[56]
Symprove^TM^ containing *Lactobacillus plantarum* NCIM 30173, *Lactobacillus rhamnosus* NCIMB 30174, and *Enterococcus faecium* NCIMB 30176	Human	↑ Proximal and distal colonic lactate ↑ Butyrate	[60]
*Lactobacillus gasseri* OLL27A	Human	↑ *Lactobacillus* spp. ↓ *Clostridium perfringens* ↑ Fecal isobutyric acid	[61]
*Lactobacillus casei*	Human	↑ Fecal butyric, propionic, and acetic acid	[63,64]

**Table 2 metabolites-13-00535-t002:** The effect of probiotics on stress response.

Probiotic	Species	Effects (↑ = Increase ↓ = Decrease)	Reference
Bifidobacterial infants	Mice	↓ Corticosterone and restore HPA axis homeostasis	[71]
*Lactobacillus helveticus* R0052 + *Bifidobacterium longum* RO175	Rats	↓ Anxiety	[73]
*L. Rhamnos*	Rats	↓ Anxiety	[74,75]
*Lactobacillus plantarum* DR7 or P8	Human	↓ Stress Symptoms	[76,80]
*Lactobacilli*, *Bifidobacteria*, *Streptocci*	Human	No change in stress symptoms	[77,78]

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
