# Peer review of "A Review of the Impact of Maternal Prenatal Stress on Offspring Microbiota and Metabolites"

_metabolites, 2023, doi:10.3390/metabo13040535_

Round 1

Reviewer 1 Report

In this review, authors discussed the impact of parental psychological stress on gut microbiota and their metabolites. The study focuses on the effect of parental psychological stress. However, sections 3 and 4 take too many parts of context to discuss the gut-brain axis and microbial metabolites, and the effects of probiotics on stress. In contrast, sections 6-9 only account for small parts of the review.

Subtitles are suggested to add to the main parts of the review to give a clear clue to audiences.  More information should be added for the main ideas of this review.

The source of the figures should be given, or the cartoons of the figures.

Other minors:

Line 15, defines SCFA in the abstract.

Change the reference style to journal type.

Author contributions, funding, and Conflicts of Interest should be included before references.

Author Response

1) In this review, authors discussed the impact of parental psychological stress on gut microbiota and their metabolites. The study focuses on the effect of parental psychological stress. However, sections 3 and 4 take too many parts of context to discuss the gut-brain axis and microbial metabolites, and the effects of probiotics on stress. In contrast, sections 6-9 only account for small parts of the review.

Subtitles are suggested to add to the main parts of the review to give a clear clue to audiences.  More information should be added for the main ideas of this review.

Your point is well taken. We expanded and reorganized sections 7-9 into one section. Not much evidence is available for section 6 discussing father’s role in offspring microbiota, therefore we highlighted this as an area that requires further investigation. 

Further, we have modified the title as suggested by another reviewer as well and focused our review on prenatal stress, microbiome changes, and probiotics on stress responses. Our intention is to only highlight potential mechanisms behind these changes and not go too deep into epigenetics since it will be beyond the scope of the topic in this review. We referred the readers to detailed reviews written on the topic.

2) The source of the figures should be given, or the cartoons of the figures.

We generated the cartoon using BiorenderTM software and have acknowledged the source in the figure legend.

Other minors:

3) Line 15, defines SCFA in the abstract.

Made this change

4) Change the reference style to journal type.

Made these changes

5) Author contributions, funding, and Conflicts of Interest should be included before references.

We included this information at the end of the document.

Reviewer 2 Report

 this work titled'the effect of parental psychological stress on the Intestinal Microbiome and microbial metabolites' reviewed evidence from human studies and animal models to discuss how maternal and paternal 13 stress can modulate the microbiome. the authors also discussed how probiotic supplementation has a pro-14 found effect on the stress response and SCFA production and how psychobiotics are emerging as 15 novel therapeutic targets. the potential molecular mechanisms by which the effects of stress are transmitted to the offspring were also discussed. I am supportive of publication.

Author Response

This work titled 'the effect of parental psychological stress on the Intestinal Microbiome and microbial metabolites' reviewed evidence from human studies and animal models to discuss how maternal and paternal 13 stress can modulate the microbiome. the authors also discussed how probiotic supplementation has a pro-14 found effect on the stress response and SCFA production and how psychobiotics are emerging as 15 novel therapeutic targets. the potential molecular mechanisms by which the effects of stress are transmitted to the offspring were also discussed. I am supportive of publication.

We thank the Reviewer for their comments

Reviewer 3 Report

In the first part of the article, the authors have made a very good review of the relationship between stress, probiotics and intestinal microbiota; however, in my opinion, they have not gone deeply enough into the relationship between parental stress and its effect on the offspring, both in the microbiota and in the development of stress and other neurological diseases in the offspring. I consider that this section should be improved, or on the contrary modify the title.

As points to improve:
The title does not accurately reflect the content of the article.
The two figures do not contribute much to the understanding of the text.

As positive points of the work it is necessary to indicate the following:

The language used is clear and fits the scientific style.

The bibliographic references are adequate and are limited to the last 10 years.

Best regards

Author Response

In the first part of the article, the authors have made a very good review of the relationship between stress, probiotics and intestinal microbiota; however, in my opinion, they have not gone deeply enough into the relationship between parental stress and its effect on the offspring, both in the microbiota and in the development of stress and other neurological diseases in the offspring. I consider that this section should be improved, or on the contrary modify the title.

We have modified the title as suggested by another reviewer as well and focused our review on prenatal stress, microbiome changes, and probiotics on stress responses. Our intention is to only highlight potential mechanisms behind these changes and not go too deep into epigenetics since it will be beyond the scope of the topic in this review. We referred the readers to detailed reviews written  on the topic.

As points to improve:
The title does not accurately reflect the content of the article.
The two figures do not contribute much to the understanding of the text.

We revised the title and Figure 2 to keep it more focused on prenatal stress and microbiome changes.

As positive points of the work it is necessary to indicate the following:

The language used is clear and fits the scientific style.

The bibliographic references are adequate and are limited to the last 10 years.

We thank the reviewer for the input and feedback to help improve the manuscript.

Reviewer 4 Report

metabolites-2182940

The manuscript of Venkata Yeramilli et al. describes The Effect of Psychological Stress During Pregnancy on The Intestinal Microbiome and Microbial Metabolites. In my opinion, it is a very significant and constantly topical issue however I am not very impressed with the way the review is composed. Below are some doubts and comments.

1. If the manuscript includes the whole spectrum of conditions of parental stress is it not better to title it: The Effect of Psychological Stress During Impregnation and Pregnancy on The Intestinal Microbiome and Microbial Metabolites?

2.      Due to the fact that in 2020 a new taxonomic classification was introduced under which 23 novel genera and a union of Lactobacillaceae and Leuconostocaceae were suggested [Zheng et al., 2020] I suggest the authors refer to the current taxonomy in their work.

3.      I see disproportions in the length of chapters. Some of them are short and not sufficiently dealing with the topic.

4.      I miss the description of how the manuscripts for this review were chosen. What was the algorithm and which databases were used?

5.     In addition, the review is prepared in a way that is not very informative. Generalizations are often supported by several cited works, unfortunately, not all of them really meet the criterion of clinical and in vivo works, and it is difficult to trace which works deal with it and what was studied in them. A tabular form of presenting the cited papers would be helpful in this respect.

In my opinion, the work needs to be systematized and put more work into its readability and merit, although the topic is very important.

Author Response

The manuscript of Venkata Yeramilli et al. describes The Effect of Psychological Stress During Pregnancy on The Intestinal Microbiome and Microbial Metabolites. In my opinion, it is a very significant and constantly topical issue however I am not very impressed with the way the review is composed. Below are some doubts and comments.

  1. If the manuscript includes the whole spectrum of conditions of parental stress is it not better to title it: The Effect of Psychological Stress During Impregnation and Pregnancy on The Intestinal Microbiome and Microbial Metabolites?

We revised the title as suggested to keep this review focused on Prenatal and Preconception stress and Microbiome changes.

  1. Due to the fact that in 2020 a new taxonomic classification was introduced under which 23 novel genera and a union of Lactobacillaceaeand Leuconostocaceae were suggested [Zheng et al., 2020] I suggest the authors refer to the current taxonomy in their work.

We included a reference to the work done by Zheng et al 2020

  1. I see disproportions in the length of chapters. Some of them are short and not sufficiently dealing with the topic.

Your point is well taken. We expanded and reorganized sections 7-9 into one section. Not much evidence is available for section 6 discussing father’s role in offspring microbiota, therefore we highlighted this as an area that requires further investigation.

We have modified the title as suggested by another reviewer as well and focused our review on prenatal stress, microbiome changes, and probiotics on stress responses. Our intention is to only highlight potential mechanisms behind these changes and not go too deep into epigenetics since it will be beyond the scope of the topic in this review. We referred the readers to detailed reviews written  on the topic.

  1. I miss the description of how the manuscripts for this review were chosen. What was the algorithm and which databases were used?

We used MeSH keywords to search for articles on PubMed

  1. In addition, the review is prepared in a way that is not very informative. Generalizations are often supported by several cited works, unfortunately, not all of them really meet the criterion of clinical and in vivo works, and it is difficult to trace which works deal with it and what was studied in them. A tabular form of presenting the cited papers would be helpful in this respect.

Your point is well taken. We compiled the evidence in into two tables to highlight the effects of probiotic treatment on the stress response

In my opinion, the work needs to be systematized and put more work into its readability and merit, although the topic is very important.

Your point is well taken. We have revised parts of the manuscript accordingly to improve readability.

Reviewer 5 Report

It is a very good study and well presented. However, there are some points need improvement

1. Absteact should have the form " introduction- materials methods-conclusion-results"

2. Initials when used should explained, for example in the abstract authors use the initials SCFA without mention it before

3. Initials should be the same when used. For example in line 94 and in line 97 authors use ECC and EC for the same thing

4. Discussion is missing. Although authors discuss in every section their findings, they should write a different section.

Author Response

It is a very good study and well presented. However, there are some points need improvement

  1. Abstract should have the form " introduction- materials methods-conclusion-results"

Your point is well taken. We followed the author guidelines recommended by the journal. Since this is not a research article we are unable to organize the information in the requested format.

  1. Initials when used should explained, for example in the abstract authors use the initials SCFA without mention it before

SCFA has been expanded during its first mention in the document.

  1. Initials should be the same when used. For example in line 94 and in line 97 authors use ECC and EC for the same thing

These changes have been addressed

  1. Discussion is missing. Although authors discuss in every section their findings, they should write a different section.

Your point is well taken. We included some discussion in each section, but we also included a new discussion section in the end to highlight main points of the review.

Round 2

Reviewer 1 Report

Authors revised the manuscript, no additional comment. 

Author Response

We thank the reviewer for their comments

Reviewer 3 Report

I consider that the manuscript has been significantly improved after incorporating the reviewers' comments.

However, in my opinion, there are some aspects that need to be improved in order to publish the article.

The title does not clearly reflect the content of the article, I think it should indicate that the authors are referring to the effects of stress during pregnancy on the offspring.

According to the authors' criteria, what is the difference between impregnation and pregnancy, so that it is necessary to include both terms in the title?

The word metabolites in line 3 is not correctly separated. It should also be indicated in the title that it is a review.

Again, in the abstract, as in the title, it is not clear whether all the effects attributed to stress on the composition of the microbiome, production of microbial-derived metabolite are in the mother or in the offspring.

Figure 1 does not provide any information as it does not indicate why probiotic administration reverses allostatic overload. I believe it should be deleted.

Yours sincerely

Author Response

I consider that the manuscript has been significantly improved after incorporating the reviewers' comments.

However, in my opinion, there are some aspects that need to be improved in order to publish the article.

  • The title does not clearly reflect the content of the article, I think it should indicate that the authors are referring to the effects of stress during pregnancy on the offspring.

We modified the title to address this point.

  • According to the authors' criteria, what is the difference between impregnation and pregnancy, so that it is necessary to include both terms in the title?

We removed the word “Impregnation” and changed the title.

  • The word metabolites in line 3 is not correctly separated. It should also be indicated in the title that it is a review.

We modified the title to include “Review” and re-positioned “Metabolites’

  • Again, in the abstract, as in the title, it is not clear whether all the effects attributed to stress on the composition of the microbiome, production of microbial-derived metabolite are in the mother or in the offspring.

We modified the title and abstract to indicate that the focus is on the offspring.

  • Figure 1 does not provide any information as it does not indicate why probiotic administration reverses allostatic overload. I believe it should be deleted.

Your suggestion is well taken. We have deleted Figure 1.

Reviewer 4 Report

In my opinion the manuscript is well revised. It is ready to be published

Author Response

(The authors gave the same response as above.)

Reviewer 5 Report

Authors made a good effort and improved thei manuscript. Although it is a review still abstract should have the form i mentioned in previous review, because it makes easier for readers to have a better view of the study

Author Response

Authors made a good effort and improved their manuscript. Although it is a review still abstract should have the form I mentioned in previous review, because it makes easier for readers to have a better view of the study

Your point is well taken. We have modified the abstract to the follow the suggested form and improve readability.

Line 42: Introduction---Introductory sentence

Lines 43-44: Materials & Methods---We used MeSH keywords to look for articles in PubMed database.

Lines 45-49: Results---We highlight results from the research articles 

Lines 50-54 : Conclusions